# A Preliminary Study on the Effect of an Intervention Based on Green Exercise on Mental Health and Physical Fitness of Adolescents

**DOI:** 10.3390/healthcare13070809

**Published:** 2025-04-03

**Authors:** Santiago Gómez-Paniagua, Carmen Galán-Arroyo, Antonio Castillo-Paredes, Jorge Rojo-Ramos

**Affiliations:** 1Promoting a Healthy Society Research Group (PHeSO), Faculty of Sport Sciences, University of Extremadura, 10003 Cáceres, Spain; sgomezpa@alumnos.unex.es (S.G.-P.); jorgerr@unex.es (J.R.-R.); 2BioẼrgon Research Group, Faculty of Sport Sciences, University of Extremadura, 10003 Cáceres, Spain; mamengalana@unex.es; 3Grupo AFySE, Investigación en Actividad Física y Salud Escolar, Escuela de Pedagogía en Educación Física, Facultad de Educación, Universidad de Las Américas, Santiago 8370040, Chile

**Keywords:** green space, life satisfaction, physical activity, physical fitness, adolescents

## Abstract

**Background**: The latest data on physical inactivity and mental health among adolescents raise concerns about the quality of life and development of young people. The expert scientific community in this field has focused its efforts on researching tools that facilitate the improvement of these variables, such as self-perceived physical condition and life satisfaction, with evidence supporting the effects of green spaces on health. **Objective**: The aim of this study is to analyze the effects of a physical activity intervention in the natural environment on life satisfaction and self-perceived physical condition in adolescents. **Methods**: For this purpose, a 12-day quasi-experimental study was carried out, consisting of nature activities (such as canyoning or canoeing) twice a day in an adolescent population that attended camps in the region. **Results**: The results showed improvements in both variables, with increasing levels of life satisfaction and self-perceived physical condition after the intervention. **Conclusions**: Physical activity in natural environments is an effective strategy to improve the physical and mental health of young people, acquiring vital importance as a protective factor against numerous psychological and social disorders. Interventions that promote physical activity in the natural environment have proven successful in improving life satisfaction and self-perceived physical condition among young people, simultaneously addressing issues of inactivity, quality of life, and healthy habits in this demographic group.

## 1. Introduction

The practice of physical activity (PA) offers an excellent opportunity to promote both personal and social values, such as respect, cooperation, social responsibility, creativity, self-knowledge, and humility, among others [1]. Panza and colleagues [2] identified a significant increase in the prevalence of adolescent anxiety disorders and depression in recent years. Currently, the prevalence of childhood and adolescent obesity is high, ranging between 8.5% and 14.8%, with a high ratio of comorbidities being expressed, in one out of three individuals, in pathologies such as depression or hypertension [3]. The World Health Organization (WHO) describes PA as any bodily movement produced by skeletal muscles that requires energy expenditure [4]. In this sense, PA is a health enhancer and an essential agent to promote healthy life habits, especially in childhood and adolescence, which are critical stages since the habits formed at these ages tend to be maintained in later stages [5]. According to research in this area, personality is developed throughout these periods [6], and promoting PA is essential to preventing issues like obesity and enhancing the physical health of this population. Thus, PA during adolescence positively impacts physical health, reduces the risk of chronic diseases such as obesity or diabetes, improves mental health, leads to higher self-esteem and better sleep quality, and helps develop social skills [7].

Moreover, Green Exercise (GE), understood as PA in green spaces [8], has numerous health benefits, both on a physical level, showing improvements in the cardiovascular and immune systems and a reduction in fatigue and drowsiness [9], and on a mental level, increasing positive emotions [10]. Natural settings are also linked to benefits for young people. Having access to green areas is linked to better mental and overall health, reduced stress [11], decreased rates of childhood depression [12], and enhanced cognitive and emotional results [13]. Additionally, green areas have been linked to reduced behavioral issues [14], hyperactivity, and issues with conduct and peer disorders [15]. In terms of physical health effects, green spaces are linked to reduced blood pressure [16], longer sleep [17], and lower rates of childhood obesity, overweight, and sedentary behavior [18]. In addition, the connection with nature during physical exercise can help reduce anxiety, depression, and stress levels and improve concentration and self-esteem [19].

In this context, life satisfaction (LS) is understood as an individual’s overall assessment of his or her quality of life [20]. Demographic characteristics such as age and gender are variables that have a limited influence on the overall LS of children and adolescents [21] and sometimes do not have a significant impact [22]. In contrast, dispositional factors seem to be more effective in predicting LS in adolescents [23] since there is evidence that variables related to self-esteem, sense of control over the environment, and social competence are linked to higher levels of LS [22]. Consequently, the concept of Self-perceived Physical Fitness (SPPF), understood as the perception that a person has of his or her own level of physical fitness [24], which can influence his or her motivation and participation in physical activities, is denoted as fundamental [25]. SPPF in adolescents may depend on several factors, such as level of PA; general health status, including the presence of chronic or mental illness; sociocultural factors; self-esteem and self-image; and previous experiences with physical exercise [26,27,28].

Currently, performing GE is associated with higher levels of psychological and general well-being, with higher values of LS compared to those who exercise in urban environments [29]. In addition, this type of PA contributes to improving SPPF in adolescence [30]. GE can contribute to greater LS and self-perception, thanks to the mental and emotional benefits of a connection with nature [31]. In this same line, it is worth mentioning the research of An and collaborators [32], which makes direct reference to the relationship between GE, LS, and SPPF in adolescents, concluding that this activity has a positive impact on both parameters and, in turn, promotes healthy lifestyle habits. However, the number of scientific papers addressing the influence of GE-based interventions on adolescent LS and SPPF is low. Most of them focus on the differences between outdoor and indoor physical activity [33], propose interventions in urban green areas [34], or examine other populations, mostly adult populations [35]. In this line, it should be noted that adolescence is a period in which both LS and SPPF suffer critical decreases, where both PA and GE are identified as efficient strategies to alleviate them [36,37]. Therefore, it is important to continue research on this topic in order to design effective strategies to promote GE among the adolescent population. Thus, the main objective of this intervention was to assess the effects of a GE-based intervention on LS and SPPF in children and adolescents in a Spanish region (Community of Extremadura). In this sense, and after reviewing the scientific literature, the intervention of GE in natural spaces will improve the levels of LS, quantified by means of the “Satisfaction with Life Scale” of adolescents belonging to the Autonomous Community of Extremadura. Likewise, this intervention will significantly improve the scores obtained in the “Visual Analog Scale of Physical Fitness Perception for Adolescents” in adolescents belonging to the Autonomous Community of Extremadura.

## 2. Materials and Methods

### 2.1. Participants

The sample of this study consisted of 420 participants (Table 1), with a slight majority of boys (51.4%, *n* = 216) compared to girls (48.6%, *n* = 204). In terms of demographic location, the majority resided in urban areas (61.2%, *n* = 257), whereas 38.8% (*n* = 163) were from rural areas. The sample was distributed over five camp shifts, with the highest number of participants in the third shift (24.5%, *n* = 103), followed by the second shift (22.9%, *n* = 96%, *n* = 963), fourth shift (21.9%, *n* = 92), first shift (16.2%, *n* = 68), and fifth shift (14.5%, *n* = 61). The mean age of the participants was 12.96 years, with a standard deviation of. 1.86, indicating moderate variability in age within the sample. This distribution of the sample by gender, demographic location, and camp shift provides a detailed basis for analysis and interpretation of the study results, reflecting a diversity of contexts and experiences among participants.

For the classification of educational centers in this study, the definition of the Cáceres Provincial Council (https://www.dip-caceres.es/) was adopted, considering those schools located in towns with a population of less than 20,000 inhabitants as rural and those in towns with more than 20,000 inhabitants as urban.

The inclusion criteria for the participants were (a) to have parental informed consent and (b) to participate in the summer camps collaborating with the research study during the summer of 2023. Likewise, participants were excluded from the study if they had a physical or mental disability that did not allow them to carry out adventure activities. The criteria for participation and exclusion were determined to match the suitability of the participants for the purposes of the study.

### 2.2. Procedure

The procedure followed to recruit participants began by contacting adventure activity companies that organized camps in natural areas. Once the agreement was made with the selected company, it was stipulated on the camp registration form that the children’s participation implied their participation in the study. Informed consent was obtained from all patients. The investigators also provided a copy of the instruments applied and a detailed explanation of the phases of the intervention so that they could transmit the information to the participants and their parents or guardians. The research team came on the day of the start of the camps to first collect the informed consent and then apply the questionnaires, explaining the items so that the participants would have no doubts about their interpretation.

Only one group was formed in which all the participants carried out the PA program, which lasted 12 days, with two main sessions: one in the morning and one in the afternoon, in which activities such as canyoning, canoeing, paddle surfing, climbing, zip-lining, hiking, river floating, caving, multi-adventure, archery, and via ferrata, with an approximate duration of three and a half hours, were carried out. Appendix A shows the first 12 sessions, corresponding to the first six days. In the following six days, the activities were repeated in a different order.

Once the PA program was implemented in the natural environment, the researchers returned to the site where the camps were held to reapply the same questionnaires and obtain post-intervention information. The design and execution of all activities were carried out by professionals of the PA and experts in outdoor adventure activities to always guarantee the well-being and physical integrity of the participants. In this sense, before each of the activities developed in the intervention, a safety talk and a briefing on the development of the session were given by the sports professionals.

According to Zurita-Cruz [38], this intervention used a quasi-experimental design with before and after measurements in a single group, where the target variables underwent two measures, one before the program was implemented and the other after.

The study was conducted in accordance with the guidelines of the Declaration of Helsinki and approved by the Ethics Committee of University of Extremadura (6/2024).

### 2.3. Instruments

Initially, a questionnaire composed of six items was designed to collect essential information about the sample, including the sex of the participants, their geographic origin, the camp shift they attended, and their age.

In addition, to evaluate the degree of satisfaction with life according to the personal perception of adolescents, the “Satisfaction with Life Scale” (SWLS), validated in Spanish by Atienza, was used [39]. This tool consists of five items designed to measure how young people value different aspects of their lives. Responses are collected on a Likert-type scale, ranging from 1 to 5, where 1 means “strongly disagree” and 5 means “strongly agree”. The authors of the scale have reported a Cronbach’s Alpha coefficient of 0.84, indicating high reliability in measuring life satisfaction in adolescents. Similarly, this tool has been used previously in GE-based interventions in the study population [40,41].

Finally, the “Visual Analog Scale of Physical Fitness Perception for Adolescents” (VAS PFA), developed by Mendoza-Muñoz and collaborators [42], was used to measure self-perception of physical fitness in adolescents. The scale includes five items that evaluate different aspects: general physical fitness, cardiorespiratory fitness, muscular strength, speed-agility, and flexibility. Each item is scored using a Likert scale from 1 to 10, where 1 means “very poor” and 10 means “very excellent”. This scale is used for the first time in a GE context since it was designed in the Physical Education classroom of public schools in Extremadura.

### 2.4. Statistical Analysis

To perform the statistical analysis of the data collected in this study, the distribution of the data was initially examined to verify compliance with the assumption of normality by means of the Kolmogorov–Smirnov test. The results indicated that this assumption was not met (*p* < 0.05), which led to the selection of nonparametric statistical tests. Therefore, the Wilcoxon test was used to compare scores before and after the intervention for each variable studied, both at the overall level and at the subgroup level in terms of gender and demographic location, establishing a significance level of *p* < 0.05. Given the nature of the data, it was considered appropriate to calculate the effect size using Rosenthal’s r for each variable to provide a quantitative measure of the impact of the intervention. The interpretation of the effect size calculated using Rosenthal’s r in this study followed the guidelines established in the literature on statistical analysis. Values of r can be classified according to their magnitude: values around 0.1 indicate a small effect, values close to 0.3 suggest a medium effect, and values of 0.5 or higher reflect a large effect [43].

The internal consistency of the psychometric scales was evaluated using Cronbach’s Alpha and McDonald’s Omega coefficients, and the values were interpreted according to Nunnally and Bernstein [44]. The maximum likelihood model (Omega ML) was used to calculate McDonald’s Omega. The data were presented using the number and percentage for the sociodemographic variables and the mean (M) and standard deviation (SD) for the scores obtained in each of the variables studied. Data analysis was performed using the statistical analysis software “Statistical Package of Social Science” (SPSS), version 23 for Media Access Control (MAC).

## 3. Results

The results of the intervention, as shown in Table 2, indicate significant improvements in LS measured by the SWLS, experiencing an improvement from 4.17 (SD = 0.73) to 4.25 (SD = 0.64), with an effect size of 0.28. In terms of gender, the intervention had a similar effect in both groups for LS, increasing significantly (*p* < 0.01) from 4.16 (SD = 0.76) to 4.33 (0.64) for males with moderate effects (r = 0.28) and for females, whose scores significantly improved (*p* < 0.01) from 3.96 (SD = 0.77) to 4.15 (0.65), also with moderate effects (r = 0.27). In the case of demographic location, the results of the subgroups show greater disparity in terms of LS. Again, both groups showed significant (*p* < 0.01) and similar increases in scores, 0.18 for urban participants and 0.20 for rural participants; however, the intervention exhibited a larger effect on rural participants (r = 0.34) compared to urban participants (r = 0.24).

Moreover, as shown in Table 3, SPPF (VAS PFA) showed a marked improvement, with mean scores increasing from 3.71 (SD = 0.70) to 3.98 (SD = 0.50) and an effect size of 0.38. In relation to the sex of the participants, men showed significant improvements (*p* < 0.01), ranging from 3.85 (SD = 0.73) to 4.07 (SD = 0.54), with a moderate effect size (r = 0.29). Similarly, the female sex exhibited significant improvements after the treatment (*p* < 0.01), but the intervention had a large effect (r = 0.48). On the other hand, participants belonging to rural settings benefited from the intervention (*p* < 0.01), with their scores going from 3.68 (SD = 0.71) to 3.93 (0.48) with a moderate effect size (r = 0.34). Meanwhile, a significant improvement (*p* < 0.01) was found in rural participants, with scores ranging from 3.76 (SD = 0.71) to 4.06 (0.51) with a large effect size (r = 0.46).

Likewise, the reliability of the instruments (internal consistency) used in this study was evaluated by calculating Cronbach’s Alpha (α) and McDonald’s Omega (ω) statistics. According to the criteria established by Nunnally and Bernstein [44], the coefficients obtained are considered satisfactory (>0.70), indicating good reliability of the instruments. It is important to note that a detailed analysis of these coefficients was performed separately for the results obtained on the scales before (pre) and after (post) the intervention (Table 4). This approach allowed an evaluation of the consistency of the instrument at different times during the study, thus ensuring the stability and reliability of the measurements throughout the intervention process.

## 4. Discussion

The aim of this work was to study the changes produced in the LS and SPPF of adolescents from Extremadura who participated in a GE-based intervention in nature. The main results obtained in this intervention indicate significant improvements in the variables studied (SPPF and LS). In terms of subgroup analysis, participants experienced similar improvements regardless of gender in LS, while participants from rural settings showed a larger effect size than their peers from urban settings. In relation to SPPF, all subgroups experienced significant improvements, with the female sex and rural settings showing a larger effect size.

The GE has a great deal of scientific evidence on its effects on health, such as sleep cycles, since exposure to natural light increases the production of melatonin, the hormone responsible for regulating the sleep cycle [45]. In addition, there is sensory stimulation due to the wide variety of visual, auditory, tactile, and olfactory stimuli to which the adolescent is exposed in this type of practice, improving mood and sense of well-being [29], acting also as a preventive agent for respiratory conditions, such as asthma [46]. In this sense, the results of this study show a significant improvement in the LS of adolescents. Similarly, research also shows a significant increase in positive affect, being more enthusiastic, attentive, and motivated [47,48], and a decrease in negative affect, feeling less nervous and upset [49], when applying GE interventions in children and/or adolescents. Nowadays, teenagers have difficulties accessing these GEs due to the lack of time because of academic and work demands, the increase in the use of technology, and the decrease in accessible GE in urban environments [50], so the developmental benefits for young people are affected.

Likewise, other research has indicated that sex cannot be considered a consistent mediator between GE and LS in the adolescent population [51]. While some publications argue for a stronger association for boys [15], others find stronger relationships for girls [52], although there is another body of literature that does not support this variation [53]. However, in the present research, the effect size for female participants was larger than for their male peers, despite the fact that recent research suggests that Spanish adolescents use natural spaces to the same extent [54]. The latest studies that have emerged in this line point to the amount of moderate and/or vigorous physical activity performed by adolescents as the fundamental factor in this mediation [55,56], agreeing with this research, since Spanish female adolescents comply with these recommendations to a lesser extent [57,58] and therefore can benefit the most from these interventions. International studies show some controversy regarding this issue, yielding different findings depending on the sociocultural context. Marquez and Long [59], who assessed the LS levels of 15-year-old adolescents in 46 nations, discovered that, generally speaking, people from rural areas had higher levels, even though there was a decline in recent years. In contrast, another study carried out in adolescents in the north of Brazil pointed to the urban population as the one with the best LS levels [60]. In this regard, Christiana et al. [61] noted that rural adolescents practiced less moderate-to-vigorous physical activity in the school environment than their urban peers and reported a lower number of displacements using active transportation, which could explain the effect sizes found in this research.

Conversely, researchers have shown that adolescents with a positive perception of physical fitness have a high probability of being physically active [62]. In addition, the self-assessment of physical fitness can be a determining factor in the motivation of young people [63]. These effects can be expanded to areas such as education and health since they promote greater physical activity in young people and an increase in motivation, which are important factors in the integral development of adolescents. The results of this study revealed an increase in SPPF after the intervention. Published studies on this subject revealed an increase in SPPF in interventions based on the practice of GE, adopting a greater perception of their own health and well-being, compared to those who exercised in urban environments [64,65]. SPPF can be affected by several sociodemographic variables, such as age, gender, and out-of-school physical/sports practice [66]; the vast majority of authors agree that gender is the most affected factor since most of the interventions conclude with a greater physical self-perception by the male sex compared to the female sex in adolescents [66,67,68,69]. Many researchers believe that these differences can be attributed to the importance of the social ideal on the body [68,69,70], where a close relationship between body image and self-esteem is observed, especially in women [67,71]. Therefore, PA and SPPF are closely related, where the increase in one of the concepts could imply the improvement of the other and greater psychological well-being [72]. In addition, SPPF does not seem to vary according to the demographic location in which the adolescents reside [73]. In this regard, body dissatisfaction has been related to higher fat percentages in Spanish-speaking adolescent populations [74], so the higher prevalence of obesity and overweight in the rural population compared to their urban peers could justify the effect size found in the present study [75,76].

In summary, GE interventions contribute to increasing LS and SPPF, thus improving quality of life and healthy habits in children and adolescents. These results could be transferred to the health field given the importance that the scientific field has found in the practice of PA in the natural environment as a means of prevention for psychological disorders due to the therapeutic effect that this activity produces in young people [77,78,79,80]. The practice of GE seems to be an effective strategy to increase the levels of SPPF and LS, with the latter understood as quality of life [48,64,65,81,82,83]. On the one hand, the importance of these results lies in the knowledge of SPPF as a tool to increase the practice of PA in young people, adopting a greater perception of their own health and well-being, thus improving their quality of life [63]. On the other hand, a higher level of LS translated into improved mental health [19]. This type of intervention could be carried out in the educational environment since this type of practice is part of the curriculum. In this sense, the possibilities are extensive; continuous visits or getaways to nearby natural areas can reduce the stress levels of adolescents. Similarly, a current trend explores the effects of natural environments through virtual reality as a great possibility in indoor environments [84]. Likewise, generating extracurricular activities that develop motor skills related to activities in the natural environment will encourage their practice in adolescents’ free time.

Regarding the limitations of the study, it is necessary to point out the demographic location since some of the participants came from rural areas and others from urban areas, so there may be cultural factors that have not been taken into account, such as PA and/or mental health levels, time of exposure to nature, or their physical self-perception [73,85,86,87]. Also, the intervention took place in the summer, so participants were more willing to carry out the activities included in the program. Likewise, the study does not have a control group and has a small sample size, so the results should be interpreted with caution. Although the published evidence suggests that the natural environment has positive effects on these variables, the results of this study should be interpreted with caution since it is not possible to determine whether green spaces are the cause of this improvement. Similarly, the prior willingness expressed by participants to engage in physical activity in nature may influence the results of the intervention. In addition, the usual PA levels of the participants were not recorded in the questionnaires, which could explain the variation in the variables studied after the intervention. As for the strengths of this study, it is one of the first interventions to study the relationship between the concepts presented, taking into account the characteristics of the study. It would be interesting to carry out the same or a similar intervention in another region to observe differences and similarities that arise and carry out more specific interventions, depending on the context where it is carried out. In this sense, to develop this type of intervention through activities in the natural environment that are more accessible in each of the study regions, facilitating the access of the adolescent population to these activities on a daily basis is important. Similarly, obtaining information on economic status, levels of physical activity, type of transport used, or social contact can be very useful mediators for research at the national level and can modify these types of interventions depending on the region in which they are developed.

## 5. Conclusions

PA in natural environment-based interventions appear to be effective in improving SPPF, LS, and mental and physical health, given the concerning rate of mental disorders and physical inactivity in the adolescent population. These interventions are crucial for preventing psychological problems and promoting healthy behaviors among young people, a population that needs support to enhance their mental and physical health because they live in an unpredictable environment.

## Figures and Tables

**Table 1 healthcare-13-00809-t001:** Sample characterization (N = 420).

Variable	Categories	N	%
Sex	Boy	216	51.4
	Girl	204	48.6
Demographic location	Rural	163	38.8
	Urban	257	61.2
Camp shift	First shift	68	16.2
	Second shift	96	22.9
	Third shift	103	24.5
	Fourth shift	92	21.9
	Fifth shift	61	14.5
Variable		M	SD
Age		12.96	1.86

N: number; %: percentage; SD: standard deviation; M: mean.

**Table 2 healthcare-13-00809-t002:** Effect of intervention on life satisfaction depending on sex and demographic location.

Intervention Group						
	N	M (SD) Pre	M (SD) Post	*p* Intra	Z	r
Life Satisfaction (SWLS)	420	4.06 (0.77)	4.25 (0.65)	<0.01	−5.67	0.277
SWLS -- Men	216	4.16 (0.76)	4.33 (0.64)	<0.01	−4.11	0.280
SWLS -- Women	204	3.96 (0.77)	4.15 (0.65)	<0.01	−3.91	0.273
SWLS -- Urban	257	4.06 (0.76)	4.24 (0.62)	<0.01	−3.87	0.241
SWLS -- Rural	163	4.06 (0.80)	4.26 (0.70)	<0.01	−4.29	0.336

Note: M = mean value; SD = standard deviation; SWLS = Satisfaction with Life Scale. r = effect size (Rosenthal’s r).

**Table 3 healthcare-13-00809-t003:** Effect of intervention on Self-perceived Physical Fitness depending on sex and demographic location.

Intervention Group						
	N	M (SD) Pre	M (SD) Post	*p* Intra	Z	r
Self-perceived Physical Fitness (VAS PFA)	420	3.71 (0.70)	3.98 (0.50)	<0.01	−7.87	0.384
VAS PFA -- Men	216	3.85 (0.73)	4.07 (0.54)	<0.01	−4.34	0.295
VAS PFA -- Women	204	3.56 (0.65)	3.88 (0.44)	<0.01	−6.85	0.478
VAS PFA -- Urban	257	3.68 (0.71)	3.93 (0.49)	<0.01	−5.44	0.339
VAS PFA -- Rural	163	3.76 (0.71)	4.06 (0.51)	<0.01	−5.85	0.458

Note: M = mean value; SD = standard deviation; VAS PFA = Visual Analog Scale of Physical Fitness Perception for Adolescents. r = effect size (Rosenthal’s r).

**Table 4 healthcare-13-00809-t004:** Reliability of the instruments applied.

	Cronbach’s Alpha	McDonald’s Omega
Self-perceived Physical Fitness (VAS PFA) pre	0.80	0.80
Self-perceived Physical Fitness (VAS PFA) post	0.71	0.71
Life Satisfaction (SWLS) pre	0.82	0.82
Life Satisfaction (SWLS) post	0.81	0.82

Note: VAS PFA = Visual Analog Scale of Physical Fitness Perception for Adolescents; SWLS = Satisfaction with Life Scale.

## Data Availability

Data will be made available upon reasonable request by the corresponding author.

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
