# Peer review of "A Preliminary Study on the Effect of an Intervention Based on Green Exercise on Mental Health and Physical Fitness of Adolescents"

_healthcare, 2025, doi:10.3390/healthcare13070809_

Round 1
Reviewer 1 Report
Comments and Suggestions for Authors
It is well known that physical activities that are performed in a natural environment have positive effects on the physical and mental health of the young population. In the current study, the authors aimed to analyze the effects of a physical activity intervention in the natural environment (green exercises) on life satisfaction and self-perceived physical condition in adolescents. I congratulate the authors on their study. My opinion on the current study is presented below.
The originality of the study and its contribution to the literature are not fully stated. The most important limitation of the study is the lack of a control group (the authors stated this in the limitations section of the study). The limitations section should also emphasize that the study was conducted in the summer. The closure of schools and the increase in air temperature will also cause an increase in outdoor activities.
In the abstract section, instead of the statistical tests you have conducted, provide information about the methodology, such as the summary of green exercises, the duration of the application, how often they are applied, and the type of activities.
Start the section as a new paragraph with the green exercise on line 43. This section first provides literature information about what green exercise is.
Author Response
It is well known that physical activities that are performed in a natural environment have positive effects on the physical and mental health of the young population. In the current study, the authors aimed to analyze the effects of a physical activity intervention in the natural environment (green exercises) on life satisfaction and self-perceived physical condition in adolescents. I congratulate the authors on their study. My opinion on the current study is presented below.
The originality of the study and its contribution to the literature are not fully stated. The most important limitation of the study is the lack of a control group (the authors stated this in the limitations section of the study). The limitations section should also emphasize that the study was conducted in the summer. The closure of schools and the increase in air temperature will also cause an increase in outdoor activities.
Response: Thank you very much for your recommendation. We have further emphasized these aspects in the limitations of the study.
In the abstract section, instead of the statistical tests you have conducted, provide information about the methodology, such as the summary of green exercises, the duration of the application, how often they are applied, and the type of activities.
Response: Thank you very much for your recommendation. This information has been substituted in the abstract of the manuscript following their indications.
Start the section as a new paragraph with the green exercise on line 43. This section first provides literature information about what green exercise is.
Response: Thank you very much for your recommendation, this issue has been corrected following your guidelines.
Reviewer 2 Report
Comments and Suggestions for Authors
We appreciate the effort put in by the authors in producing this article.
We have a few questions and observations:
- Lines 33-35 refer to the years 2015-2016. Are there no more recent studies? It would be good to add them.
- The participants are the 420 teenagers. Were they constantly present at physical education classes in the period prior to inclusion in this project/study? Do they practice any sports in their free time?
- What relevance does the different number in each group have for the study? How were the groups formed?
- Did the camps take place at the same time or successively?
- The authors of the article tell us that the teenagers agreed to participate in these camps voluntarily. Did this fact influence the results obtained?
- Were the teenagers familiar with the types of activities that took place in the camp or with similar activities? (the activities are, from our point of view, quite difficult, even involving risks!) What was the level of preparation of the teenagers before the experiment?
- Line 251 talks about possible cultural factors that were not taken into account. What would they be? To what extent could they have influenced the research results?
Author Response
- Lines 33-35 refer to the years 2015-2016. Are there no more recent studies? It would be good to add them.
Response: Thank you very much for your recommendation. These lines have been replaced by last year's worldwide data, also pointing out the most common comorbidities including mental health problems.
- The participants are the 420 teenagers. Were they constantly present at physical education classes in the period prior to inclusion in this project/study? Do they practice any sports in their free time?
Response: We fully understand your recommendation and your concerns about the PA levels of the adolescents who participated in the intervention. Unfortunately, these data were not collected in this particular intervention. Following your recommendation, it has been proposed in the research group to include these questions in future research given the clear association between adolescent PA levels and mental health/self-perception of physical fitness. This issue has also been included in the limitations of the study.
- What relevance does the different number in each group have for the study? How were the groups formed?
Response: We really understand your concerns. The research team contacted several companies that developed camps focused on these activities in the region during the summer, and since only one of them wanted to collaborate, all the adolescents who attended each of the shifts were taken into account in order to recruit the largest possible sample. All the activities developed in each of the shifts were the same, but the number of participants in each of them depended mainly on the willingness of the parents for the dates of celebration of the shifts. Our intention was to have been able to randomize the sample and to have had a control group, but this was not possible despite our will, mainly due to the number of participants.
- Did the camps take place at the same time or successively?
Response: The camp shifts lasted 15 days and were held in succession, but the research group ensured that the activities were carried out in a timely manner.
- The authors of the article tell us that the teenagers agreed to participate in these camps voluntarily. Did this fact influence the results obtained?
Response: We fully understand their reasoning, however, the participants or their legal guardians were the ones who expressed in advance the will for them to participate in summer camps with a major PA component in nature. The agreement with the company simply lies in substituting the PA modalities they normally carry out for those proposed by the research team. As you rightly point out, this preference for PA in nature may influence the results, so it has been included in the limitations of the study.
- Were the teenagers familiar with the types of activities that took place in the camp or with similar activities? (the activities are, from our point of view, quite difficult, even involving risks!) What was the level of preparation of the teenagers before the experiment?
Response: Thank you very much for your recommendation, it is true that we have not exposed this information in the “Procedure” section. Before each of the activities developed in the intervention, a safety talk and a briefing on the development of the activities was given by the sports professionals. This issue has been clarified in the aforementioned section.
- Line 251 talks about possible cultural factors that were not taken into account. What would they be? To what extent could they have influenced the research results?
Response: Thank you very much for your recommendation, we understand the confusion it creates for the reader. Therefore, we have mentioned in the same line several cultural factors dependent on the demographic location of the participants that may affect the results along with the relevant scientific literature.
Reviewer 3 Report
Comments and Suggestions for Authors
Introduction
- Clarify the Research Gap: The introduction provides a solid background, but it could be clearer about what’s missing in current research. Highlight why this study is necessary and how it fills a gap in understanding.
- Stronger Justification for Adolescents: You mention that adolescence is a critical stage for forming habits, but you could dive deeper into why this group is particularly at risk and how GE (Green Exercise) specifically benefits them at this stage of life.
- Improve Reference Use: It feels like some references (like [13]) are being used to support multiple points in the same paragraph, which can make the argument seem a bit cluttered. It might help to break the points up for better clarity and flow.
- Refine the Hypothesis: The hypothesis is currently broad. A more focused statement on what exactly you expect GE to impact—specifically in terms of life satisfaction (LS) and self-perceived physical fitness (SPPF)—would strengthen the study. Also, explaining how these outcomes will be measured in more detail would help.
Methodology
- Research Design: The quasi-experimental design is solid for tracking changes, but adding a control group could really help clarify if the intervention is the key factor driving the results. This would make your conclusions more robust.
- Participants & Sample Distribution: The sample seems balanced, but the fact that 61.2% of participants are urban could potentially influence the results. It might be interesting to look at how urban versus rural participants respond differently to the intervention, if at all.
- Measurement Tools & Reliability: The Satisfaction with Life Scale (SWLS) and Visual Analog Scale of Physical Fitness Perception (VAS PFA) are good tools, but it would help to mention if they’ve been used in similar contexts, like camps or outdoor settings, to ensure they’re appropriate for your population.
- Data Analysis & Statistical Approach: Nonparametric tests are a smart choice given the data distribution. It would be useful to dig a little deeper into the data by looking at different subgroups—like gender or urban vs. rural participants—just to see if these factors play a role. Additionally, adding confidence intervals for effect sizes would give readers a clearer sense of the intervention's impact.
Results
- Control Group: Including a control group would provide a stronger foundation for your conclusions about causality.
- Urban vs. Rural Differences: Consider looking more closely at whether urban and rural participants respond differently to the intervention.
- Psychometric Tool Justification: Clarify why the psychometric tools you’ve chosen are appropriate for this study.
- Subgroup Breakdown: Breaking down results by subgroups (e.g., gender or location) would add depth to your findings and make them more comprehensive.
Discussion
- Contextualize Effect Sizes: The effect sizes are moderate, but it would be helpful to explain what this means practically for the readers. How might these changes play out in real-world settings?
- Subgroup Analysis: A deeper look at different variables like age, gender, or camp shift could provide more insights. Did certain groups benefit more than others? This could add nuance to your findings.
- Reliability Results: The reliability scores are strong, but offering a comparison with similar studies could help readers gauge their consistency and relevance.
- Confidence Intervals for Effect Sizes: Adding confidence intervals would give readers more certainty about the reliability of your results.
- Explore Cultural Influences: Since your participants come from both urban and rural areas, it might be worth considering how cultural and environmental factors could influence your results. A deeper dive into this could help explain any discrepancies.
- Further Explanation of Socio-Demographic Factors: You mention that gender, age, and physical activity play a role in SPPF, but it would be helpful to explain how these variables interacted during the study. It could clarify why some groups showed more significant improvements.
- Recommendations for Future Research: Replicating this study in other regions is a great idea, but it would be even stronger if you provided more specific suggestions on how future research could address cultural and regional differences.
- Practical Applications: The results are promising, but it would be helpful to offer more practical advice on how these interventions could be used in schools or communities. Clear recommendations on implementing GE interventions would make your study’s findings more actionable.
Author Response
Introduction
- Clarify the Research Gap: The introduction provides a solid background, but it could be clearer about what’s missing in current research. Highlight why this study is necessary and how it fills a gap in understanding.
Response: Thank you very much for your recommendation. It is true that the gap in the literature covering this research is not sufficiently clear. Therefore, these aspects have been included in the introduction to clarify the reader's understanding.
- Stronger Justification for Adolescents: You mention that adolescence is a critical stage for forming habits, but you could dive deeper into why this group is particularly at risk and how GE (Green Exercise) specifically benefits them at this stage of life.
Response: Thank you very much for your recommendation. The importance of LS and SPPF in adolescence and their relationships to mental health and physical fitness have been described. Likewise, the possibilities and benefits of GE in adolescence have been emphasized.
- Improve Reference Use: It feels like some references (like [13]) are being used to support multiple points in the same paragraph, which can make the argument seem a bit cluttered. It might help to break the points up for better clarity and flow.
Response: Thank you very much for your recommendation. This important issue has been corrected throughout the article.
- Refine the Hypothesis: The hypothesis is currently broad. A more focused statement on what exactly you expect GE to impact—specifically in terms of life satisfaction (LS) and self-perceived physical fitness (SPPF)—would strengthen the study. Also, explaining how these outcomes will be measured in more detail would help.
Response: Thank you very much for your recommendation. The hypotheses have been separated and clearly explained according to your observations.
Methodology
- Research Design: The quasi-experimental design is solid for tracking changes, but adding a control group could really help clarify if the intervention is the key factor driving the results. This would make your conclusions more robust.
Response: Thank you very much for your recommendation. We fully understand the importance you point out about the control group. The intention of the research team was to carry out an experimental and randomized design, however, only one company agreed to collaborate with us in this research, so we were forced to include all participants in the intervention group in order to collect as much data as possible. It has been proposed in the research group, following your recommendation, to repeat the intervention with the collaboration of two or more companies to carry out a randomized study. For clarification purposes, we have introduced the term “Preliminary Study” in the title.
- Participants & Sample Distribution: The sample seems balanced, but the fact that 61.2% of participants are urban could potentially influence the results. It might be interesting to look at how urban versus rural participants respond differently to the intervention, if at all.
Response: Thank you very much for your recommendation. Following this recommendation, the data analysis has been carried out again taking into account the subgroups mentioned above.
- Measurement Tools & Reliability: The Satisfaction with Life Scale (SWLS) and Visual Analog Scale of Physical Fitness Perception (VAS PFA) are good tools, but it would help to mention if they’ve been used in similar contexts, like camps or outdoor settings, to ensure they’re appropriate for your population.
Response: Thank you for your recommendation. The SWLS scale has been widely used in PA contexts and interventions in nature, as it has been around for more than 30 years as stated in the “Instruments” section following your suggestion. On the other hand, the VAS PFA has been validated in adolescents from Extremadura, more specifically in the context of the PE classroom, this study being the first approach to the natural environment. This has also been included in the previously mentioned section.
- Data Analysis & Statistical Approach: Nonparametric tests are a smart choice given the data distribution. It would be useful to dig a little deeper into the data by looking at different subgroups—like gender or urban vs. rural participants—just to see if these factors play a role. Additionally, adding confidence intervals for effect sizes would give readers a clearer sense of the intervention's impact.
Response: Thank you very much for your recommendation. In this sense, the statistical analysis has been expanded, contemplating the different subgroups mentioned in regard to the intervention. As for the confidence intervals for the effect size, the research team has not found any scientific document that facilitates their calculation. To our humble understanding, these are associated with normal distributions (https://doi.org/10.4097/kjae.2016.69.6.555), which would be possible in other statistics such as Cohen's D, but not in the present study. Please, if you know a reliable method to calculate them in the Rosental r, it would be of great help if you could provide it to us.
Results
- Control Group: Including a control group would provide a stronger foundation for your conclusions about causality.
Response: Thank you very much for your recommendation. We fully understand the importance you point out about the control group. The intention of the research team was to carry out an experimental and randomized design, however, only one company agreed to collaborate with us in this research, so we were forced to include all participants in the intervention group in order to collect as much data as possible. It has been proposed in the research group, following your recommendation, to repeat the intervention with the collaboration of two or more companies to carry out a randomized study. For clarification purposes, we have introduced the term “Preliminary Study” in the title.
- Urban vs. Rural Differences: Consider looking more closely at whether urban and rural participants respond differently to the intervention.
Response: Thank you very much for your recommendation. On your advice the results of the study have been extended to the sex and demographic location subgroups.
- Psychometric Tool Justification: Clarify why the psychometric tools you’ve chosen are appropriate for this study.
Response: Thank you very much for your recommendation. Following your advice, the explanation about the suitability of the questionnaires has been extended in the subsection “Instruments” and the cut-off values in the results have been clarified to make the reader understand the good results.
- Subgroup Breakdown: Breaking down results by subgroups (e.g., gender or location) would add depth to your findings and make them more comprehensive.
Response: Thank you very much for your recommendation. On your advice the results of the study have been extended to the sex and demographic location subgroups.
Discussion
- Contextualize Effect Sizes: The effect sizes are moderate, but it would be helpful to explain what this means practically for the readers. How might these changes play out in real-world settings?
Response: Thank you very much for your recommendation. Throughout the discussion, where the differences in the subgroup analyses have been discussed, the effect sizes found in the present investigation have been contextualized and justified.
- Subgroup Analysis: A deeper look at different variables like age, gender, or camp shift could provide more insights. Did certain groups benefit more than others? This could add nuance to your findings.
Response: Thank you very much for your recommendation. Variables such as age, gender and demographic location have been addressed throughout the discussion. As for the shifts of the camps, this issue was subject to the preference of the legal guardians and the participants, as only one company collaborated in the study, the research team was forced to recruit all participants for all shifts. What we can say for sure is that all activities were carried out in the same way regardless of the shift.
- Reliability Results: The reliability scores are strong, but offering a comparison with similar studies could help readers gauge their consistency and relevance.
Response: Thank you very much for your recommendation. Unfortunately, the studies proposing interventions in nature and/or green spaces using these scales do not provide reliability values of the scales with the study populations, they only refer to the reliability values of the scales in the initial validation studies (see https://doi.org/10.1080/09603123.2019.1577368 or https://doi.org/10.1080/17439760.2023.2209538). Our intention was to show that the scales were appropriate for these interventions, both at the level of study population and temporal stability.
- Confidence Intervals for Effect Sizes: Adding confidence intervals would give readers more certainty about the reliability of your results.
Response: Thank you very much for your recommendation. As for the confidence intervals for the effect size, the research team has not found any scientific document that facilitates their calculation. To our humble understanding, these are associated with normal distributions (https://doi.org/10.4097/kjae.2016.69.6.555), which would be possible in other statistics such as Cohen's D, but not in the present study. Please, if you know a reliable method to calculate them in the Rosental r, it would be of great help if you could provide it to us.
- Explore Cultural Influences: Since your participants come from both urban and rural areas, it might be worth considering how cultural and environmental factors could influence your results. A deeper dive into this could help explain any discrepancies.
Response: Thank you very much for your recommendation. Having addressed the differences between subgroups as you recommended, information has been added to the discussion that addresses this issue as well as the sociocultural factors that may affect the results.
- Further Explanation of Socio-Demographic Factors: You mention that gender, age, and physical activity play a role in SPPF, but it would be helpful to explain how these variables interacted during the study. It could clarify why some groups showed more significant improvements.
Response: Thank you for your recommendation, this information has been added throughout the discussion.
- Recommendations for Future Research: Replicating this study in other regions is a great idea, but it would be even stronger if you provided more specific suggestions on how future research could address cultural and regional differences.
Response: Thank you very much for your recommendation. Instructions have been added at the end of the Discussion in reference to possible future research in the area.
- Practical Applications: The results are promising, but it would be helpful to offer more practical advice on how these interventions could be used in schools or communities. Clear recommendations on implementing GE interventions would make your study’s findings more actionable.
Response: Thank you very much for your recommendation. We have added concise practical application information at the end of the discussion.
Round 2
Reviewer 1 Report
Comments and Suggestions for Authors
I congratulate the authors for the changes and additions made in line with the opinions and suggestions of the reviewer. Although the additions and corrections made are sufficient, the tables should be rearranged according to APA format.
Author Response
I congratulate the authors for the changes and additions made in line with the opinions and suggestions of the reviewer. Although the additions and corrections made are sufficient, the tables should be rearranged according to APA format.
Response: Thank you very much for your recommendations throughout the review process. Following the latter, we have adapted the formatting of the tables to the APA standards
Reviewer 3 Report
Comments and Suggestions for Authors
Thanks to the authors for their important fixes; this version appears ready for publication. However, I still believe there are some gaps, including the small sample size, the lack of a control group, and a few other factors that might be considered limitations of the study.
Author Response
Thanks to the authors for their important fixes; this version appears ready for publication. However, I still believe there are some gaps, including the small sample size, the lack of a control group, and a few other factors that might be considered limitations of the study.
Response: Thank you very much for your recommendations throughout the review process. Following the latter, these factors have been mentioned in the limitations of the study, see lines 322 to 335.